# Research and design of an intelligent fish tank system

**Shaohua Fu** *, Wenjing Xing, Juncheng Wu, Jiangang Chen, Shuangjiao Liu

All authors work on Big Data and Internet of Things School, Chongqing Vocational Institute of Engineering, Chongqing, China

* fsh1108@cqvie.edu.cn

## Abstract

In order to improve the hardware configuration and interaction mode of the fish tank system and realize the diversification of client functions, the purpose of real-time remote monitoring and management is achieved. A set of IoT intelligent fish tank system composed of sensor unit, signal processing unit and wireless transmission unit was designed. The system improves the algorithm of the data collected by the sensor, and proposes an improved first-order lag average filtering algorithm. The system uses composite collection information, intelligent processing, chart data analysis and other methods to transmit the processed data to the cloud server through the WIFI communication module. An APP is designed on the remote monitoring and control end, and a visual data interface of the smart fish tank is made, and the user can modify the environmental parameters conducive to the biological survival inside the fish tank through the APP, it brings great convenience to the family fish tank, and the test shows that the system network is stable and fast in response, and the overall purpose of the intelligent fish tank system is achieved.

## I. Introduction

With the development of economy and the gradual improvement of the national income level, people's living standards have gradually improved, many people take fish farming as a supplement to daily life, and fish tanks have also become an indispensable tool for fish farming. However, most of the traditional fish tanks are semi-automatic products, and they generally exist, such as single function, remote monitoring, and inability to set biological living environment parameters [1–3], these problems make it impossible for users to raise fish efficiently, accurately and scientifically, so we have designed a smart fish tank system based on the Internet of Things to solve the above problems. Zhicheng Liu proposed a smart fish tank with multiple functions in the design of intelligent fish tank based on single-chip microcomputer [4], the intelligent fish tank is based on a single-chip microcomputer as the control center, the tank is designed around the basic operation of the daily maintenance tank, combined with sensor technology, automatic adjustment of the environment in the tank, however, the single-chip microcomputer has small memory and slow response speed. To solve the response speed, Xuan Peng in the design and research of intelligent fish tank based on STM32 single-chip

**Competing interests:** The authors have declared that no competing interests exist.

microcomputer [5], the use of STM32 embedded microcontrollers improves the response speed and increases the memory, but the network transmission speed is slow. Yuhao Guo and his team proposed a home smart fish tank design based on STM32 [6], the system has the functions of intelligent water circulation inside and outside the fish tank, automatic feeding temperature regulation, intelligent reminder of newborn small fish, and system emergency power supply, which can effectively help users easily and scientifically raise ornamental fish, but does not consider the mobile terminal optimization display. Shunke Liang and his team proposed a 4G-based IoT fish tank system design [7], optimized the scheme, upgraded the network connection by using 4G, and added a motion monitoring APP, which can realize remote monitoring and remote control, making it more convenient for users to raise fish, but the APP display interface is simple and there is no real-time environmental parameter monitoring display. Qiujing Zhang and his team proposed an intelligent fish tank culture system design based on the Internet of Things [8], which senses water quality changes through sensors, and interacts with users through communication modules to reflect water quality information in real time, thereby greatly improving the efficiency and intelligence of fish tanks, but not effectively optimizing the hardware network, increasing the system workload.

In addition, there are some other studies, such as the intelligent fish tank culture system based on STM32 [9], the intelligent fish tank system based on the Internet of Things [10] and the design of the intelligent fish tank remote control system [11], all of which have put forward some research on the design of the intelligent fish tank. However, as a complex network containing multiple elements such as users, objects, and network facilities, the intelligent fish tank frequently interacts between people and things, between things, and between things and networks, and there are problems that its comprehensive performance needs to be improved.

With the development of the country's new infrastructure, most public places and homes have been installed with WIFI, and 4G or 5G in public places have also achieved comprehensive coverage [12]. In this paper, through the combination of Internet of Things technology and mobile communication network, an integrated control system based on embedded intelligent fish tank is designed, the system uses sensors to collect temperature, PH value, water level and other environmental data in the fish tank, uses differential communication to deploy sensor networks, uses STM32F103 processor as the main control module to process data, and adopts responsive network layout, uses WiFi or 4G to upload data to the cloud, realizes the automatic control of fish tank environmental parameters through real-time display through mobile phone APP or desktop computer. The system can also realize the functions of oxygenation, enhanced lighting, automatic water change, automatic feeding, and realize remote monitoring and control of the fish tank, the system has the advantages of high anti-interference and strong practicability, providing users with a better experience and meeting the requirements of users to intelligently manage fish tanks.

## II. The overall structure of intelligent fish tank remote monitoring system

The intelligent fish tank remote monitoring system uses the Internet of Things and intelligent equipment to monitor the whole process of fish survival. DS18B20 temperature sensor, BH1750 light intensity sensor, and water quality PH sensor arranged in a fish tank, Real-time collection of water temperature, light intensity, water PH value living environment parameters, suitable water temperature, light intensity and PH value of water bring a comfortable living environment to fish and improve the survival rate of fish. The main control module includes the circuit board, STM32F103 processor, bus communication interface, the data at the sensor acquisition end is processed by the algorithm written by the main control chip

STM32F103,send AT(Attention) commands through the serial port to connect to WIFI and send environmental data to the user through the MQTT(Message Queuing Telemetry Transport) server, customers can remotely control and adjust the control actuator of the fish tank at any time, computer terminal and mobile phone APP, Operate the feeding module, water purification module, and oxygenation module to change the environmental state, the overall system framework is shown in Fig 1.

## III. system hardware design

### A. Main controller module design

This paper uses STM32F103 chip as the main controller module chip of the system. The microcontroller core is ARM32—bit Cortex-M3 processor, with the main frequency of 72Mhz [13], small size and low power consumption. It has 64KB FLASH program memory, 20KB RAM and 12—bit ADC, which have a total of 12 channels, 37 general I / O ports, 4 16—bit timers, 2 * IIC, 2 * SPI, 3 * USART, 1 * CAN, working voltage of 2V – 3.6V and working temperature of − 40 ˚C– 85 ˚C. These characteristics of STM32F103 microcontroller also make it widely used in embedded systems. In the main controller module design keys, serial ports, network ports and other peripherals to achieve other control functions, in addition, the circuit board also designed a digital signal and analog signal input terminal to receive a variety of detected data to achieve comprehensive monitoring of fish tank environmental data. In order to make the system more versatile, the voltage conversion function is integrated on the circuit board. simply connects the main control board to the power supply, and the specific main controller module circuit board is shown in Fig 2.

### B. Data acquisition sensor module

The data acquisition terminal mainly collects the environmental information of the fish tank. The data acquisition module of water temperature, water PH value and light intensity is designed in this project.

Water temperature sensor module: The water temperature will directly affect the growth of fish in the fish tank. Different types of fish have different water temperature requirements. The project uses collecting and controlling water temperature to create good living conditions for fish. In this regard, the integrated sensor with DS18B20 protective shell is the most common digital temperature sensor in this project. The sensor has strong waterproof performance and ensures that it will not be damaged by water in long-term use. In the test, the sensor is placed in the bath. When the test data are stable, the measured and recorded data can be used. The schematic diagram of the temperature measurement peripheral circuit is designed as Fig 3.

PH value of water quality: the survival of fish has certain range requirements for PH value of water quality. The project uses high-precision PH sensor for water quality detection. PH7 indicates that the concentration of H+ is less than that of OH−, so the solution is alkaline. So the smaller the pH [14], the stronger the acidity of the solution; the higher the PH is, the stronger the alkalinity of the solution is. The main control board has the corresponding amplification processing circuit to test the PH of soil water faster and more accurately. When installed, the probe is placed in the water to fully contact with the water. Do not place the entire sensor in water. The schematic diagram of water PH measurement circuit is designed as Fig 4.

Light intensity: the greater the sunlight, the faster the evaporation of oxygen concentration in water. This project uses light intensity sensor to collect light intensity to realize automatic oxygen supply. BH1750 light intensity sensor is selected, and the sensor needs to be placed outside to collect the sunlight intensity during installation. Light intensity acquisition interface circuit schematic diagram as Fig 5.

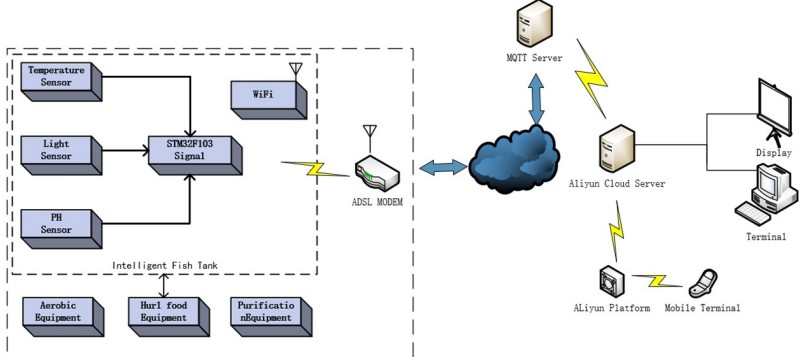

**Fig 1. Overall system structure diagram.**

Communication unit design. After the data of the sensor acquisition end are processed by the algorithm in the MCU of the embedded main controller module, the AT(Attention) instruction is sent through the serial port to connect WIFI and the environmental data are sent through the MQTT(Message Queuing Telemetry Transport) server. The test needs to connect WiFi at home. During installation, the antenna needs to be leaked out.

## C. Control the executing agency

The project control actuator is divided into feeding module, water purification module, oxygen module. The control actuator is mainly used to improve the environmental state of the fish tank.

Feeding module. Feeding is an essential function of intelligent fish tank. This project drives the mechanical structure by motor. When testing, the feeding mode needs to be calibrated to get more accurate feeding. Control module circuit is divided into two parts: stepper motor circuit, relay control circuit, which relay control water pump and gas pump. Fig 6 is the stepper motor drive circuit [15], Fig 7 is the relay control circuit.

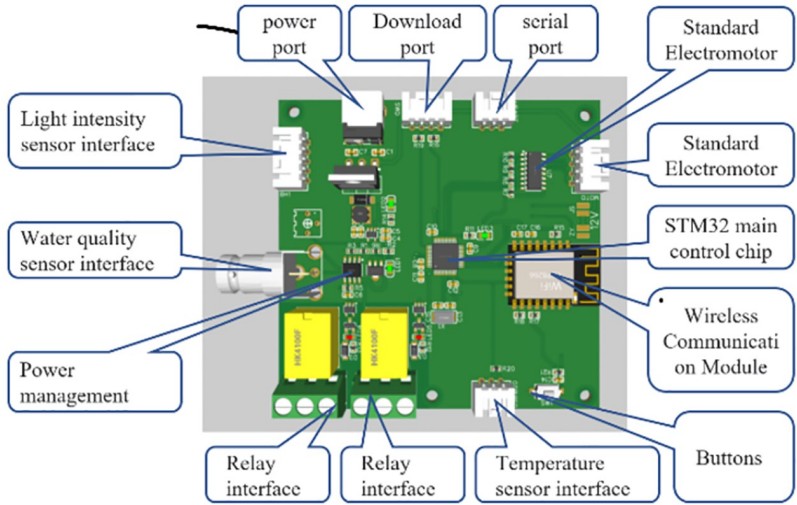

**Fig 2. Main controller module circuit board.**

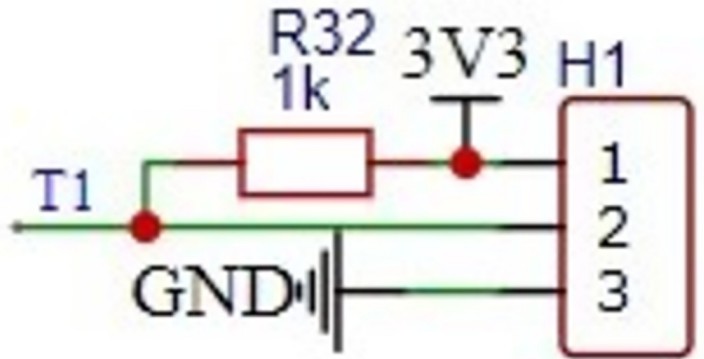

**Fig 3. Schematic diagram of peripheral circuit for temperature measurement.**

Water purification module: Fish survival also has certain requirements for water quality, driven by the system detection module for water purification. The water quality in this project was purified by the filter, and the corresponding mechanical structure was designed to extract and purify the water in the cylinder. During installation, the water pipe should be placed in water with a filter.

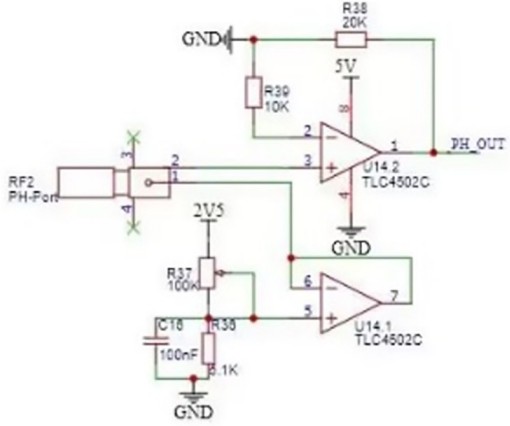

**Fig 4. The schematic diagram of water PH measurement circuit.**

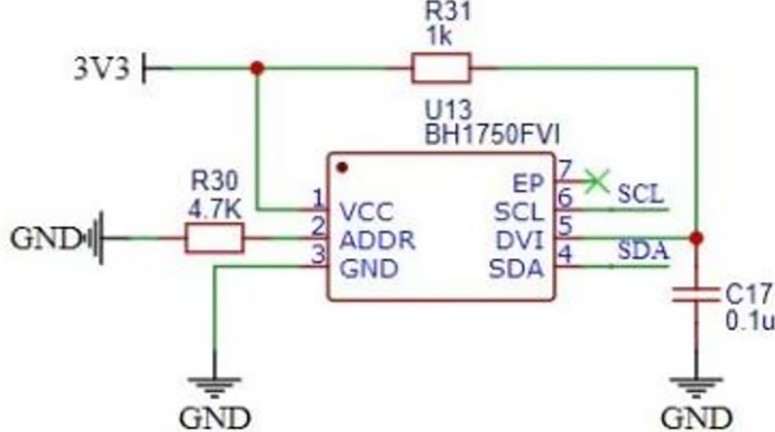

**Fig 5. Light intensity interface circuit schematic diagram.**

Oxygenation module: The project uses miniature gas pump as oxygen source after hypoxia in fish tank. During installation, the aeration head needs to be placed in water for oxygen.

### D. Summary of hardware selection

We made a hardware selection table as shown in Table 1.

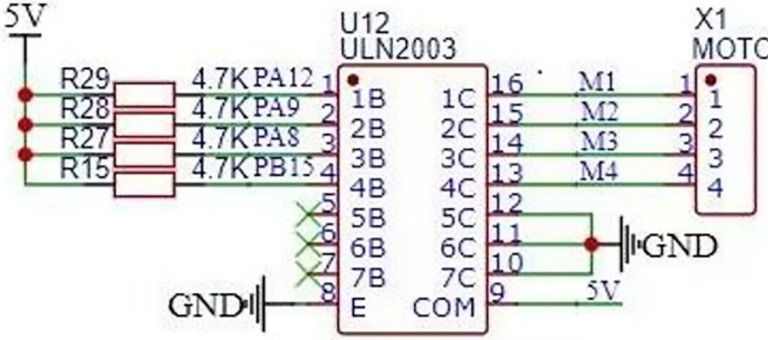

**Fig 6. Stepping motor driving circuit.**

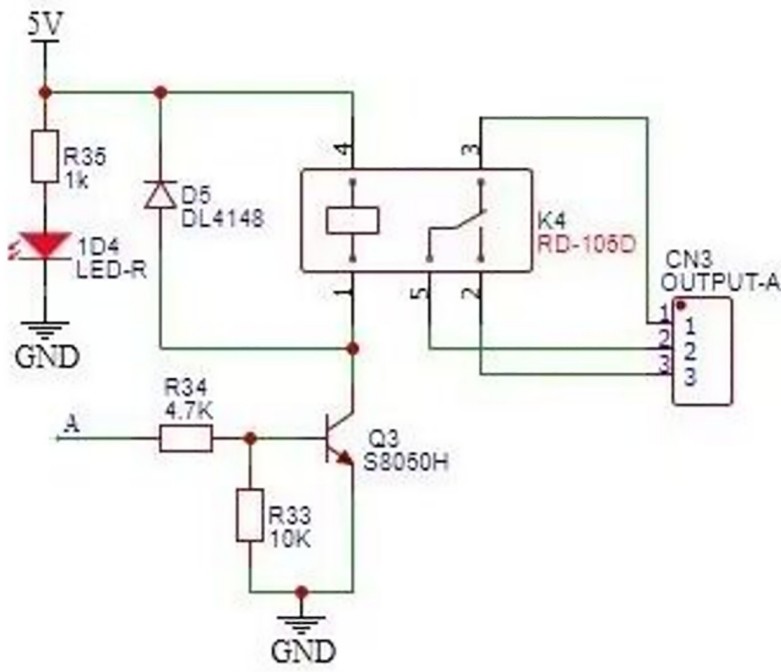

**Fig 7. Relay control circuit.**

## IV. The software design of intelligent fish cylinder system

### A. Overall system framework

In order to meet the requirements of the overall development of the system, the C language and Keil uVision5 compiler will be used for embedded program development and debugging. The program flow of this project is to first initialize the system and configure the system's sensors and wireless communication modules, After initialization, connect to the server, and report the environmental data collected by the sensor to the Alibaba Cloud platform after the

**Table 1. Hardware selection.**

| Hardware selection | Voltage(V) | Current(A) | Model |
|---|---|---|---|
| Master control | 3.3 | 0.3 | STM32F103C8T6 |
| Stepper motor | 6 | 1 | HT3705 |
| Oxygen pump | 12 | 1.5 | Uv_u1 |
| water pump | 12 | 1.2 | HQB-2000 |
| PH sensor | 5 | 0.5 | RMD-ISSF-5 |
| Temperature sensor | 3.3 | 0.3 | DHT11 |
| Network transmission chip | 3.3 | 0.3 | ESP8266 |
| Light sensor | 3.3 | 0.3 | BH1750 |

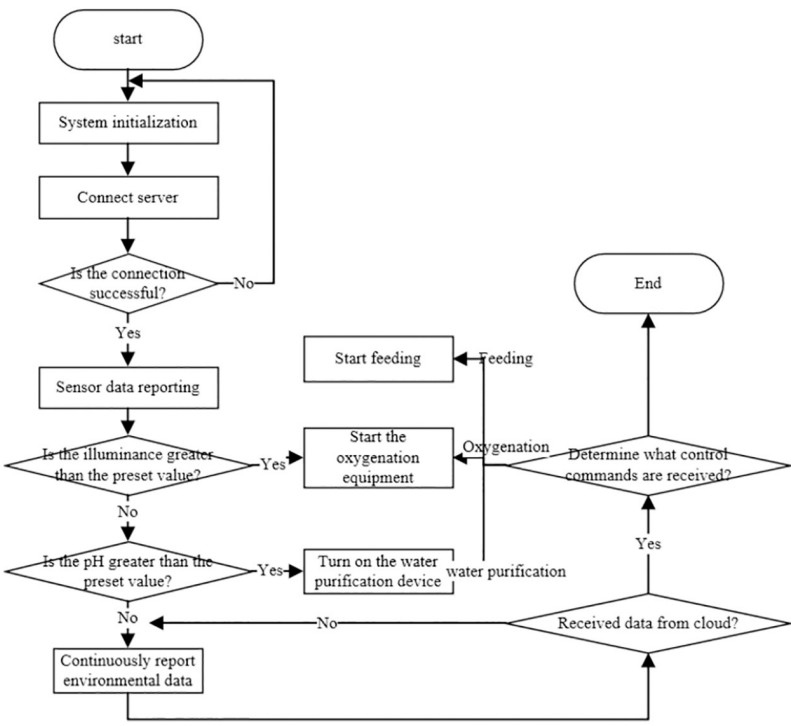

**Fig 8. Overall framework of the system.**

connection is successful, At the control end, when the light value exceeds the preset value, the oxygenation operation will be performed, and when the PH value is greater than the preset value, the water purification system will be turned on [16]. The overall system framework process is shown in Fig 8.

## B. Environmental detection framework

Sensor drive system is based on temperature sensors, light sensors, PH sensors to collect the surrounding environmental data for processing. When the temperature, light intensity, PH value exceeds the preset value, it will open oxygen, water purification and other functions to maintain the ideal value of the environment. The sensor drive flow chart is shown in Fig 9.

## C. Wireless communication drive framework

Wireless communication uses WIFI module, which can realize remote control only by connecting WIFI. Driver is used to drive WIFI module to access the Internet for remote control. Firstly, initialize the module, and report the sensor data after accessing the server. The cloud receives the reported data and forwards it to the APP. When receiving the instructions issued by the cloud to the microcontroller [17], the corresponding oxygen, feeding, water purification and other operations are performed. The wireless communication driver flowchart is shown in Fig 10.

## D. Filtering algorithm

In terms of data acquisition and processing, through the above analysis, this design combines the first-order lag filtering method with the median average filtering method, and proposes an

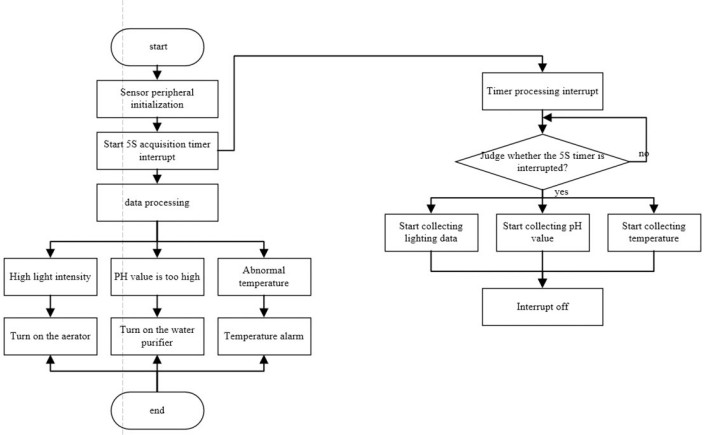

**Fig 9. Environment detection framework.**

improved first-order lag average filtering method. The algorithm flowchart is shown in Fig 11. After STM32 reads the nth sampling value of the sensor output.

Then, it is sent into the queue for the improved first-order lag low-pass filtering processing. The nth obtained value is multiplied by the filtering coefficient $\frac{a}{2}$, and the n-1th obtained value is multiplied by the filtering coefficient $(1 - \frac{a}{2})$. The final value Y1n after the result processing is

$$Y_{1n} = \frac{a}{2}X_n(1 - \frac{a}{2})\,Y_{n-1}.\tag{1}$$

Where Y1n is the output value of this filtering, Xn is the output value of this filtering, Yn−1 is

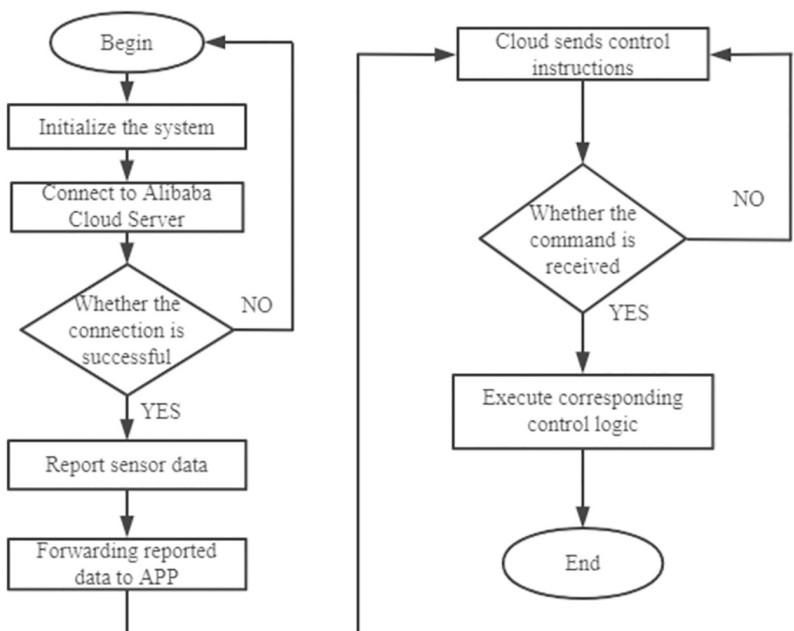

**Fig 10. Wireless communication driver framework.**

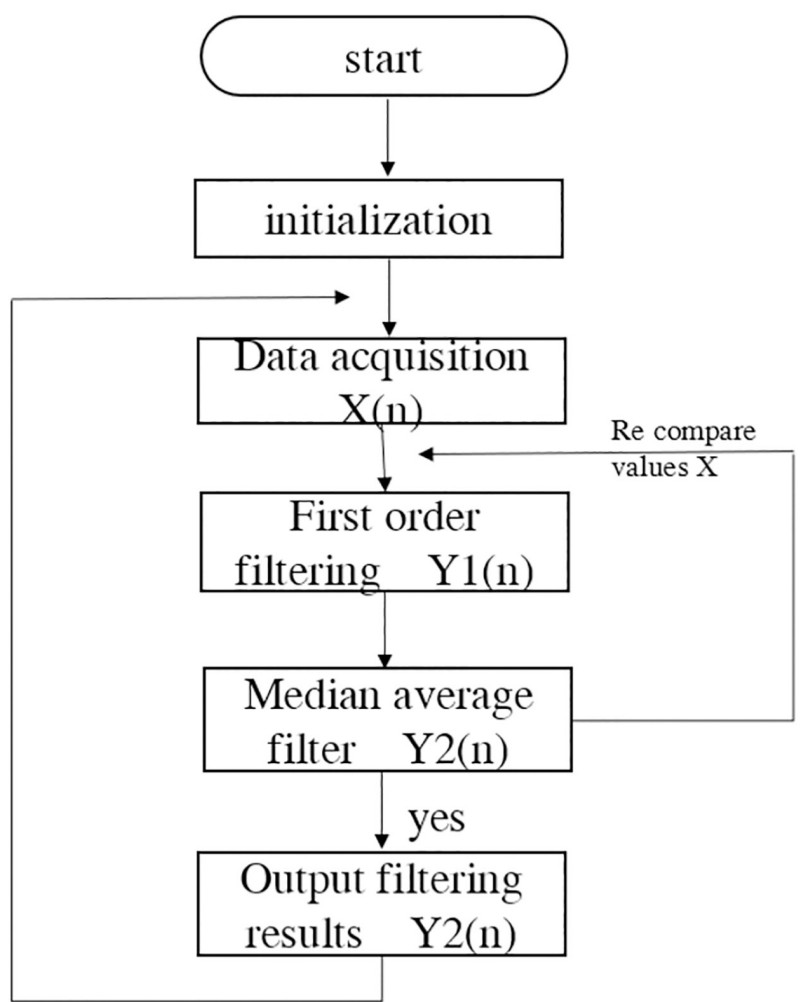

**Fig 11. Flow chart of improved first-order lag average filtering algorithm.**

the last filtered output value, α is the filtering factor, and the low-pass filtering algorithm α used in this design is 0.1.

The data processed by the improved first-order lag low-pass filter are processed by median average filtering, that is, N1 measured values are continuously collected, and the N1 values are regarded as a one-dimensional array. A minimum and a maximum in the array are removed, and then the arithmetic average of the remaining (N-2) data is calculated as the latest filtering output, and updated to the historical sampling average. The sample value processed by median average filtering is the final output value Y2(n)

$$Y_{2(n)} = \frac{1}{N_i - 2} \sum_{i=0}^{N_1 - 1} \left( Y_{n-i} - Y_{(n-i)max} - Y_{(n-i)min} \right) \tag{2}$$

In the formula, $Y_{n-i}$ is the output value of the first (n-i) sampling after the first-order low-pass filtering. $Y_{(n-i)min}$ and $Y_{(n-i)min}$ min are the maximum and minimum values of the sampling values in the array, respectively.

The improved first-order lag average filtering algorithm combines the advantages of the first-order lag filtering method and the median average filtering method, which can eliminate

**Table 2. Algorithm Comparison.**

| Number \ Name | First-order hysteresis filtering | average filtering | First-order lagged average filtering algorithm |
|---|---|---|---|
| 1 | 0.8 | 0.75 | 0.76 |
| 2 | 1.8 | 1.85 | 1.8 |
| 3 | 3.4 | 3.2 | 3.1 |
| 4 | 4 | 4.2 | 4.2 |

the pulse interference of the gas production signal caused by the flow and other related actions in the fish tank, and suppress the data error caused by the periodic interference, and improve the accuracy of the sampling environment value test in the fish tank. We used the sample data collected in real time to make an algorithm comparison table and algorithm curve diagram, as shown in Table 2 and Figs 12–14.

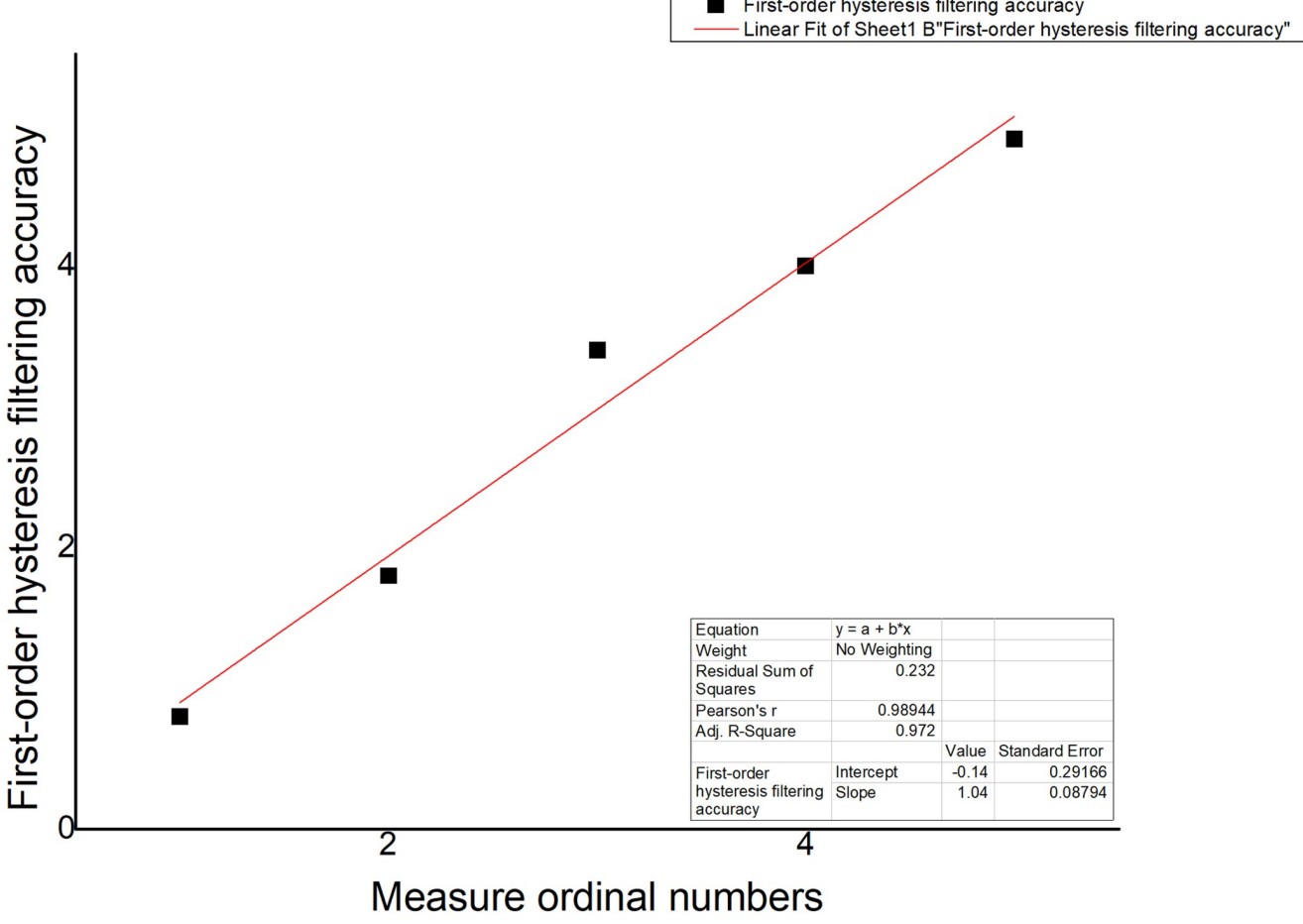

**Fig 12. First-order lag averaging filtering algorithm.**

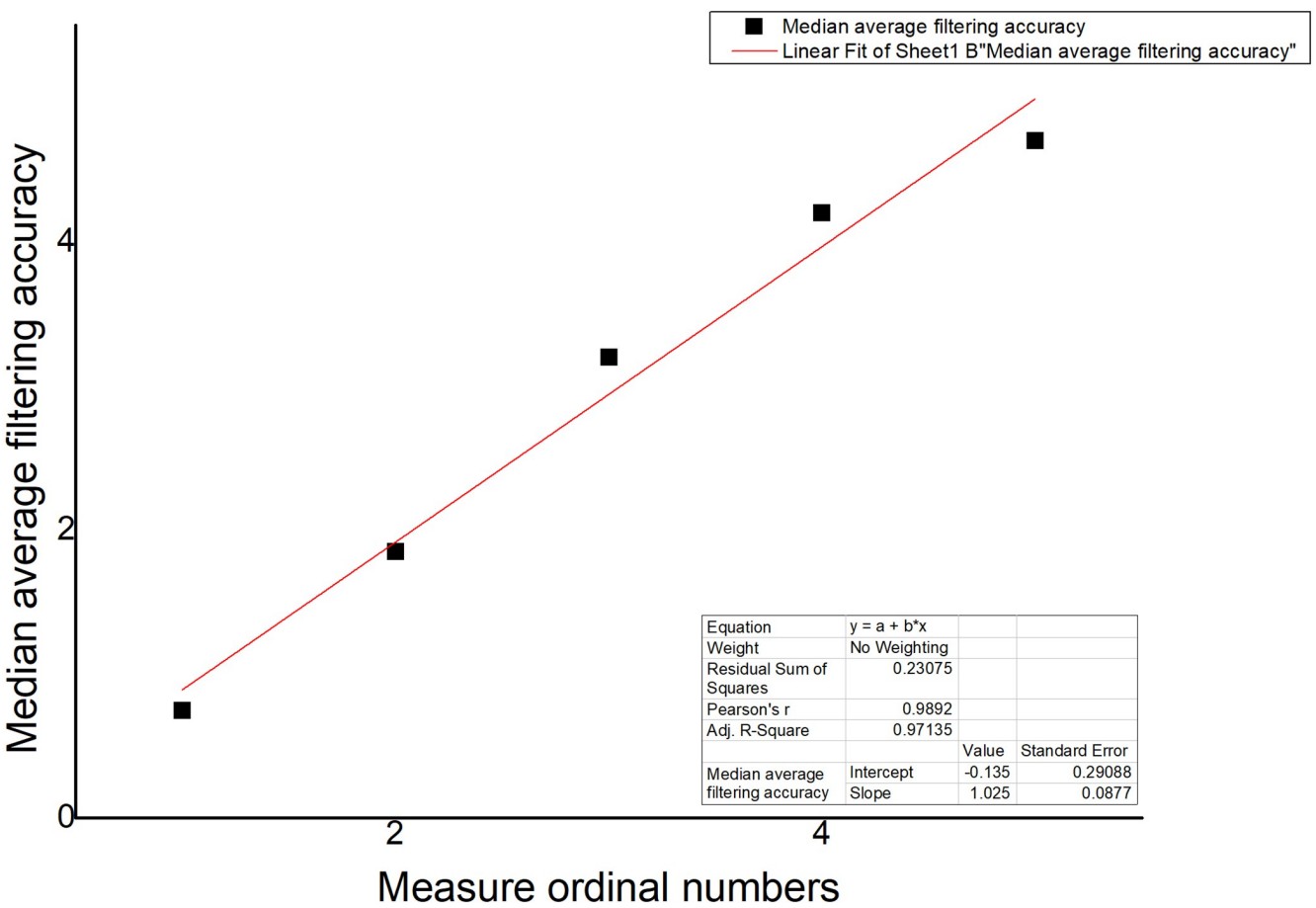

**Fig 13. Median average filtering algorithm.**

## V. Experimental analysis and testing

System software testing includes: System operation stability test, data acquisition accuracy and stability test, data transmission stability test, control instruction execution stability test, remote instruction receiving accuracy and stability test. Among them, the stability of PH device, oxygen concentration sensor, and temperature sensor detection module is tested by comparing actual sampling data with reference data; The control module adopts the comparison of data delivery times and actual reaction times to test the accuracy and performance of data; The communication module adopts two-way command data communication, and compares the accuracy of packet capture to detect its stability. Make the smart fish tank as Fig 15, mobile phone APP interface as Fig 16.

During the operation of the system, the monitoring module will always collect data, and long-term operation of the system may affect the stability and accuracy of this module. Stability is shown in Fig 17, where the actual data collected by the detection module is compared with the reference data in order to judge the stability of the detection module in high strength and long-term use, and to achieve the role of the test detection module.

When the system is running, the control module, as the instruction execution module, needs to maintain high module stability to ensure the normal operation of the whole system. Data acquisition stability is shown in Fig 18. Here, the background is used to issue instructions

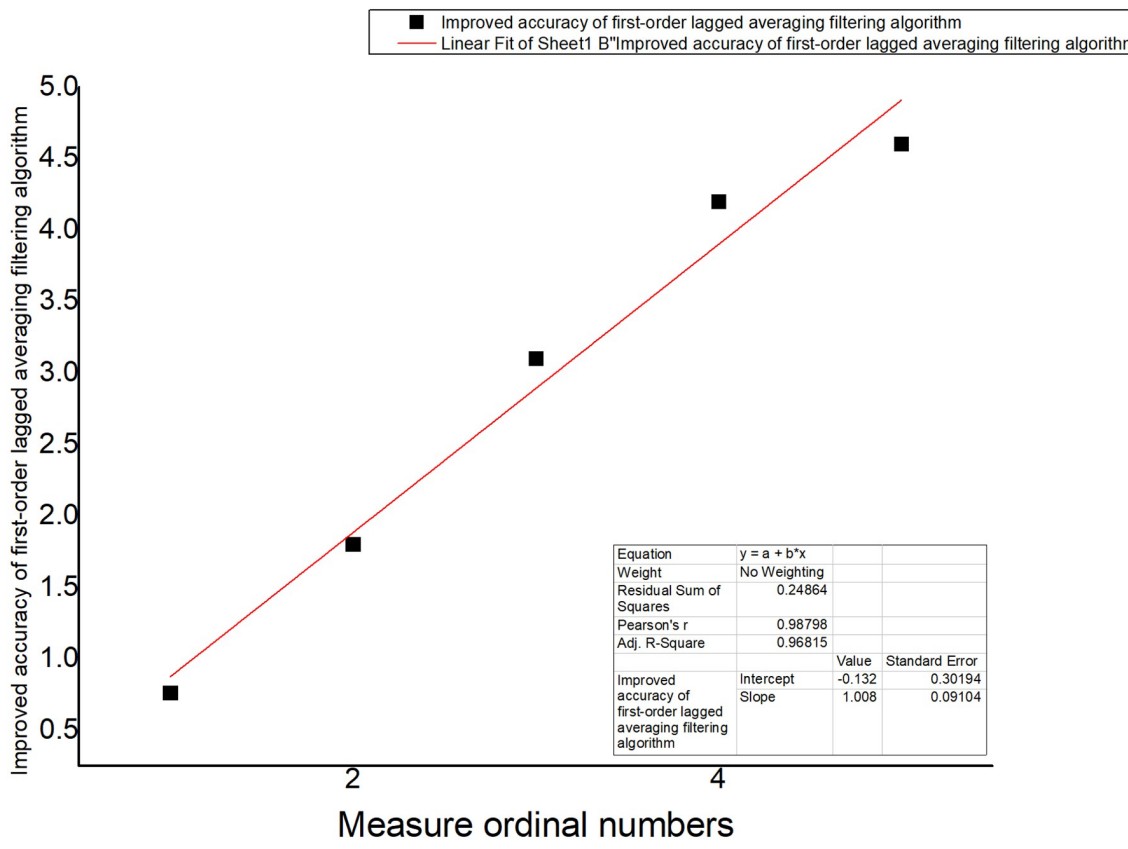

**Fig 14. Improved first-order lag averaging filtering algorithm.**

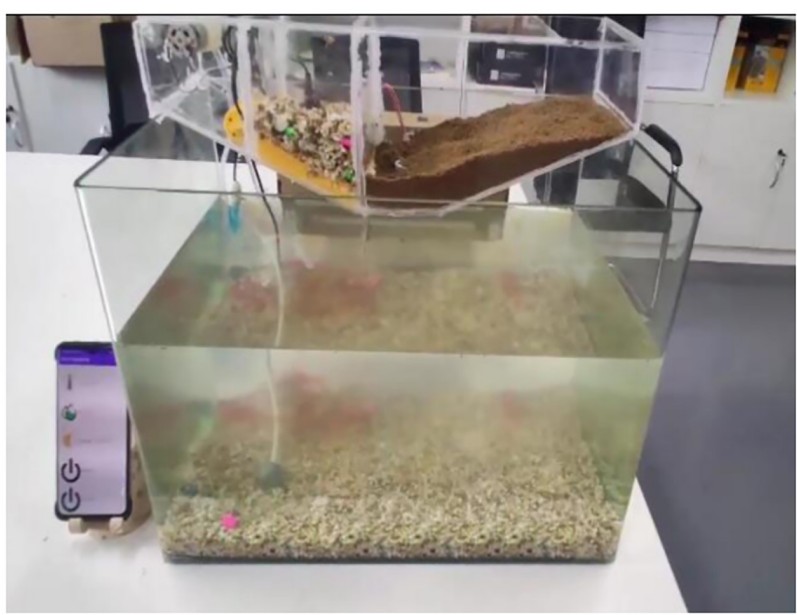

**Fig 15. Smart fish tank.**

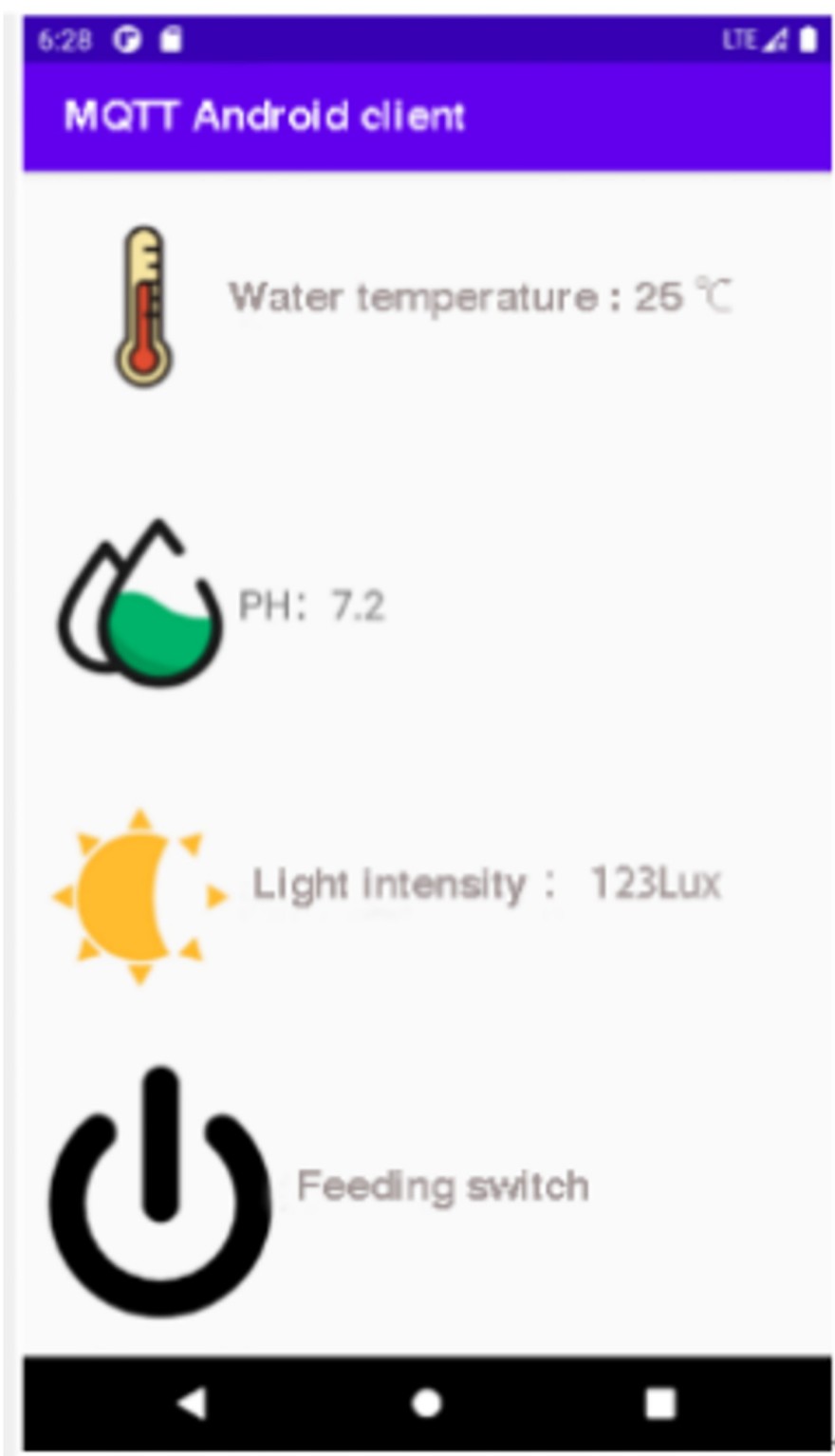

**Fig 16. Mobile phone APP interface.**

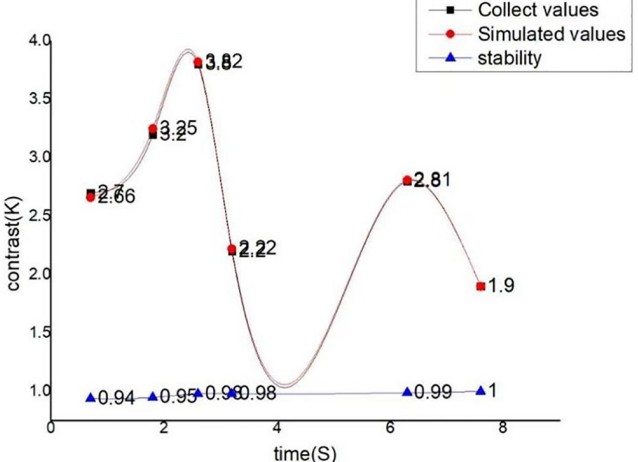

**Fig 17. The stability comparison.**

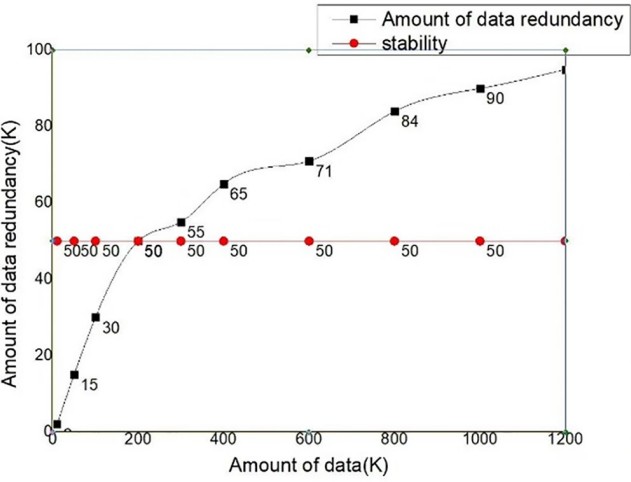

**Fig 18. Data acquisition stability.**

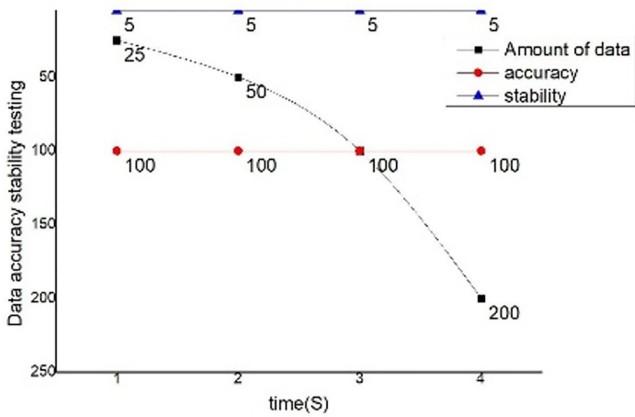

**Fig 19. Comparison diagram of data accuracy and stability.**

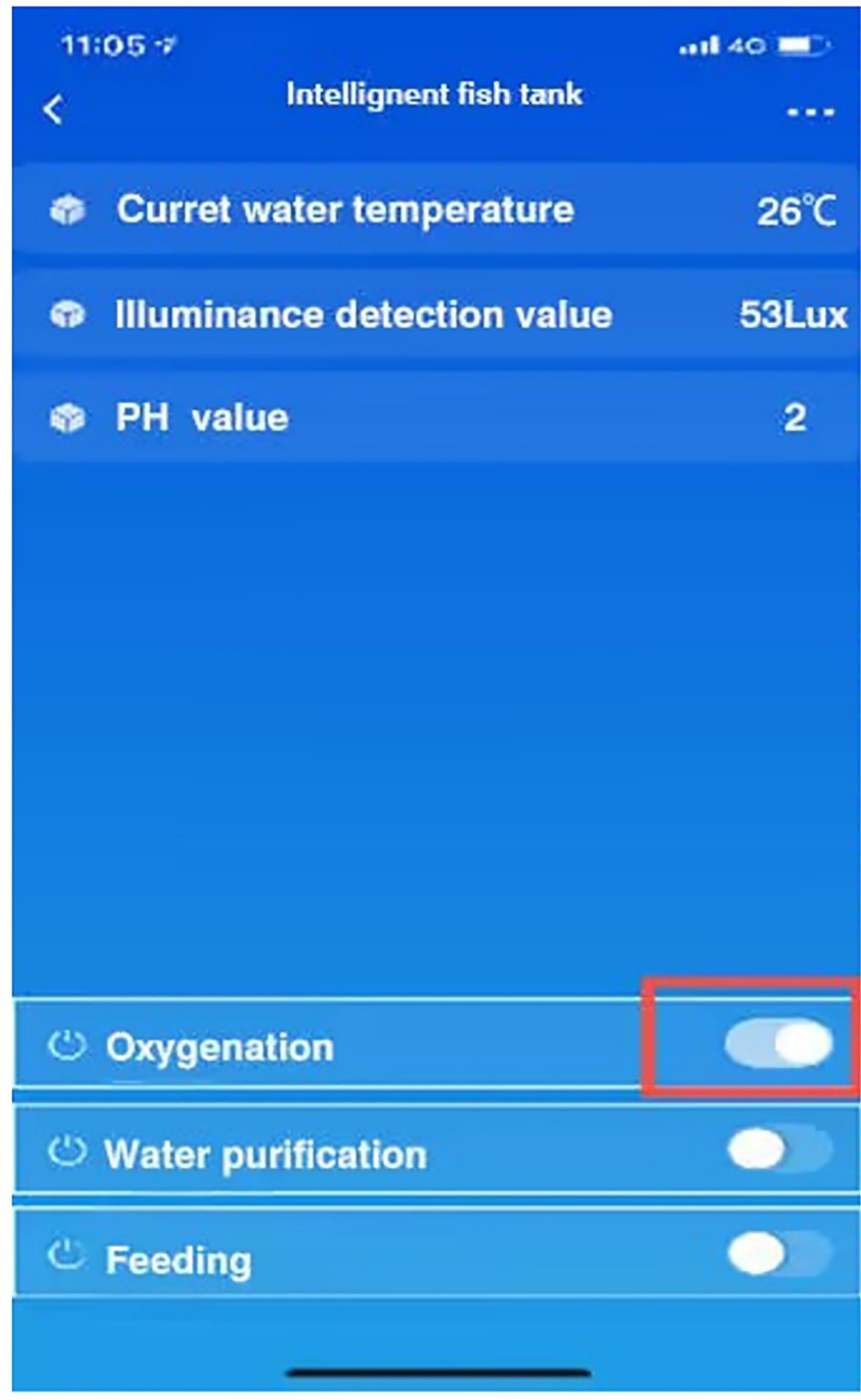

**Fig 20. Directive for oxygen enrichment.**

in a short period of time, and compare whether the issued instruction set and the accepted instruction set correspond correctly to judge whether the control module runs correctly and whether the control module instructions are executed correctly or not, so as to achieve the purpose of testing the stability of the control module.

When the instruction is issued and the data is uploaded, the communication module must ensure the accuracy and stability of the instruction and data, the background is used to issue instructions in a short time, and upload them at the same time to judge whether the issued instruction set and the accepted instruction set correspond correctly, so as to achieve the purpose of testing the stability of the communication module. Comparison diagram of data accuracy and stability are shown in Fig 19.

In the running time of the system, the control module will make the corresponding response to achieve the corresponding effect after receiving the instructions. This test is to further test the overall operation of the system after the mobile terminal issued the opening pump instructions, whether the system can fully execute the instructions, check whether the oxygen enrichment function reaches the expectation, the oxygen enrichment issued instructions schematic diagram and control diagram are shown in Figs 20 and 21. Simulation commands are issued, data is recorded, and expected results are observe.

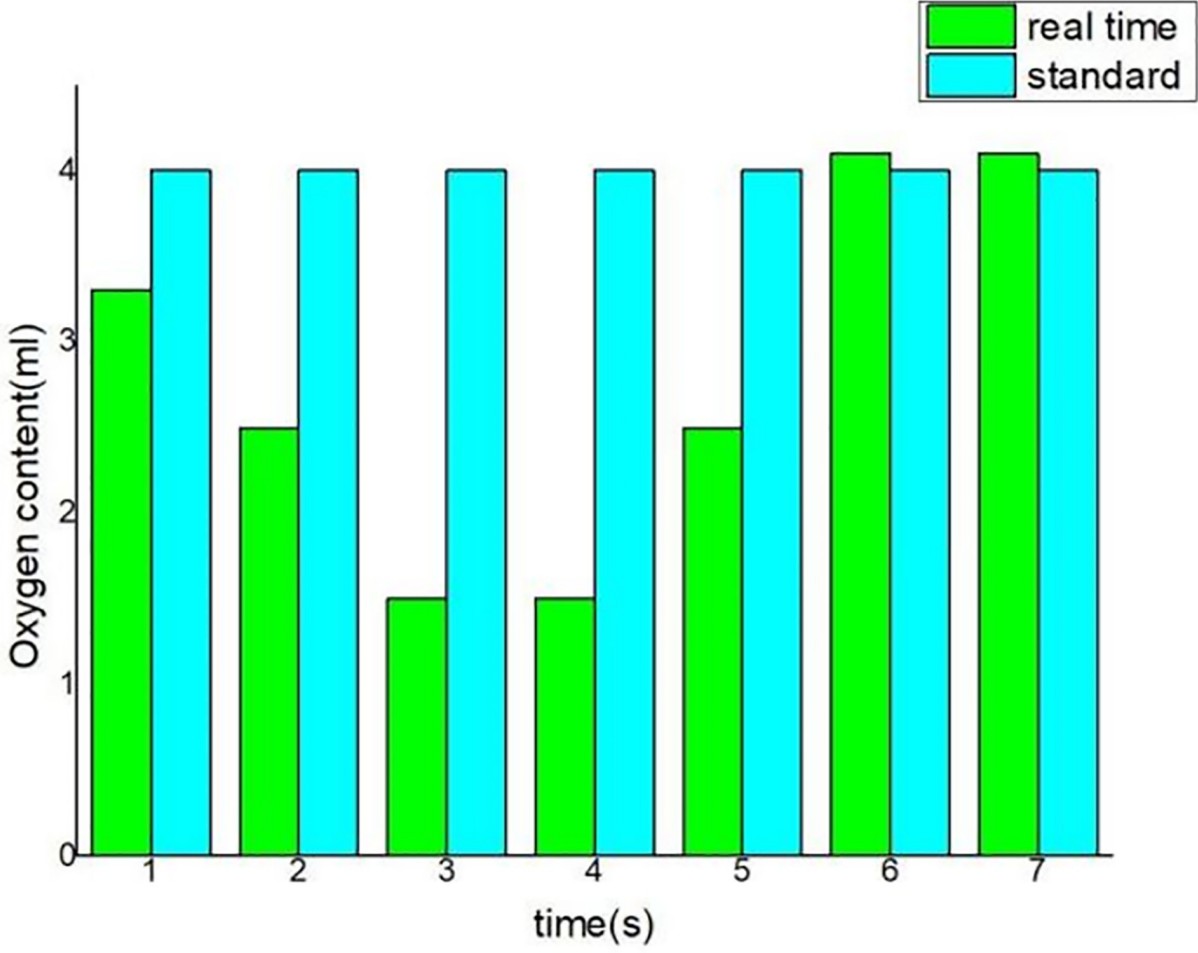

**Fig 21. Comparison chart of oxygen content change.**

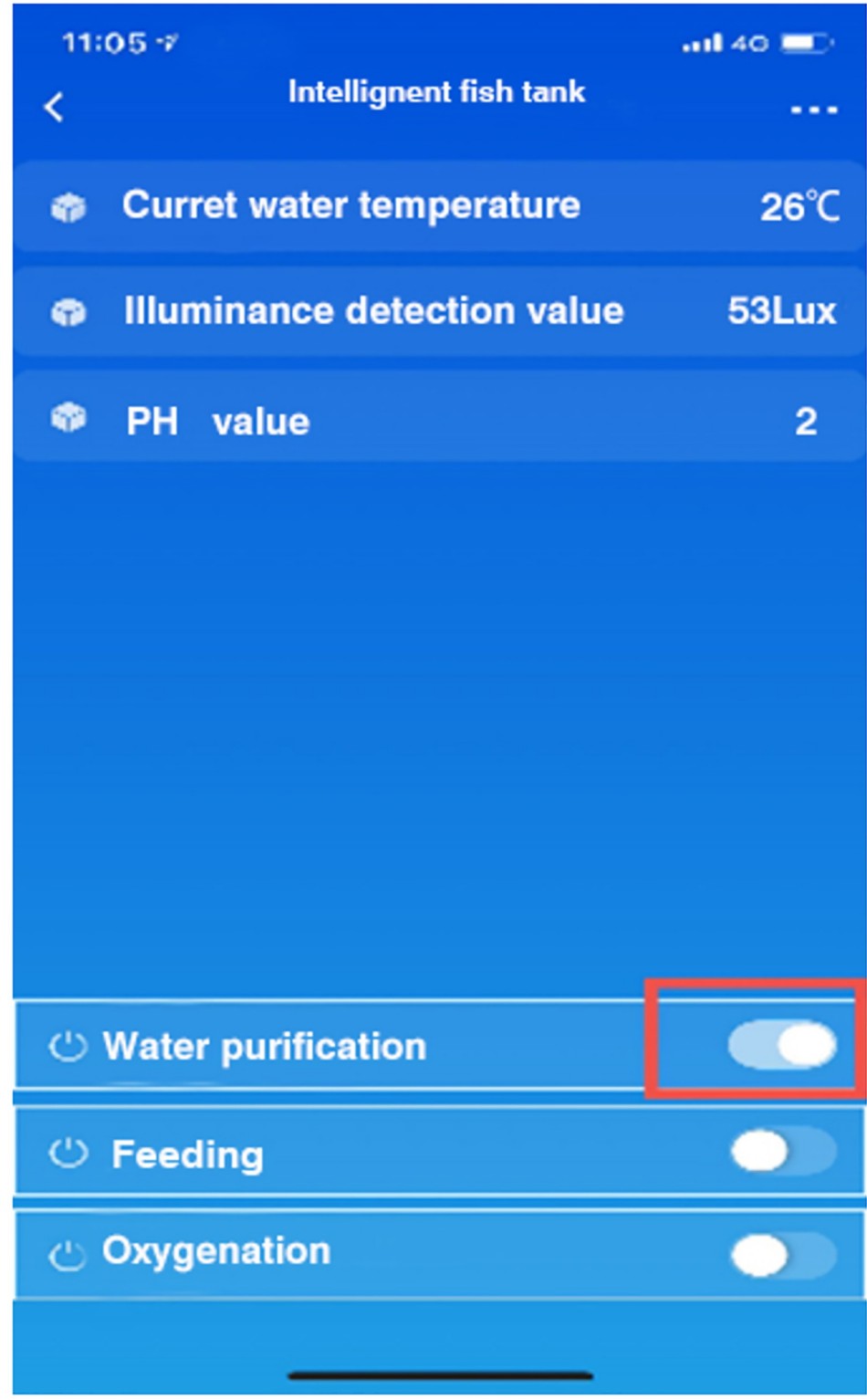

**Fig 22. Water purification instruction issued.**

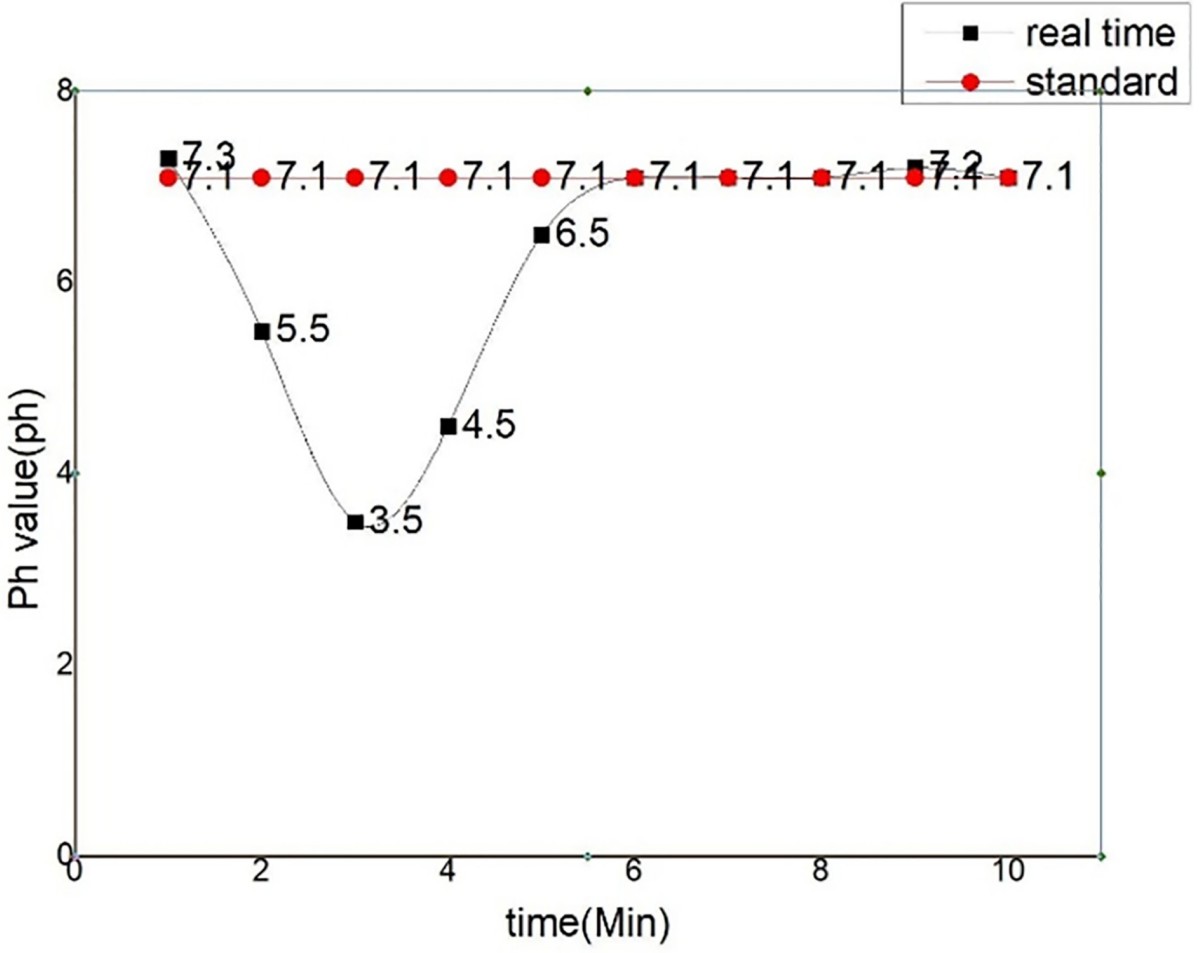

**Fig 23. Contrast chart of water quality change.**

In the running time of the system, the control module will make the corresponding response to achieve the corresponding effect after receiving the instructions. This test is to further test the overall operation of the system, the mobile terminal issues the opening pump instructions, whether the system can fully execute the instructions and see whether the water purification function reaches the expectation. The distribution chart and control chart of water purification instructions are shown in Figs 22 and 23. Simulation commands are issued, data is recorded, and expected results are observed.

During the operation time of the system, the control module will make corresponding reactions after receiving the instructions to achieve the corresponding effect. This test is to further test whether the system can fully implement the instructions and see whether the feeding function meets the expectations after the mobile terminal sends out the opening motor instructions after the overall operation of the system. The distribution chart and control chart of feeding instructions are shown in Figs 24 and 25. Simulation commands are issued, data is recorded, and expected results are observed.

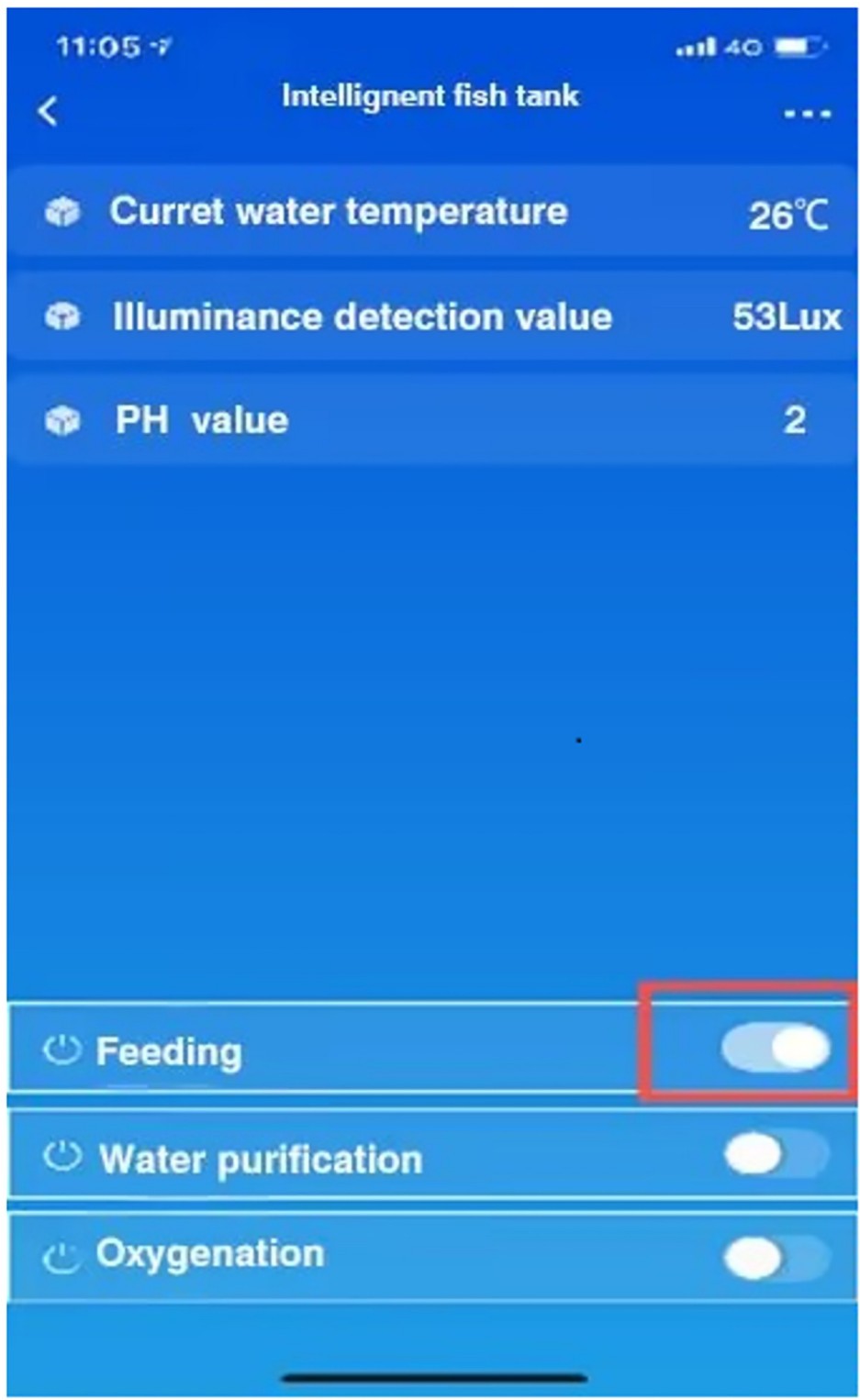

**Fig 24. Feeding the Layout Instruction.**

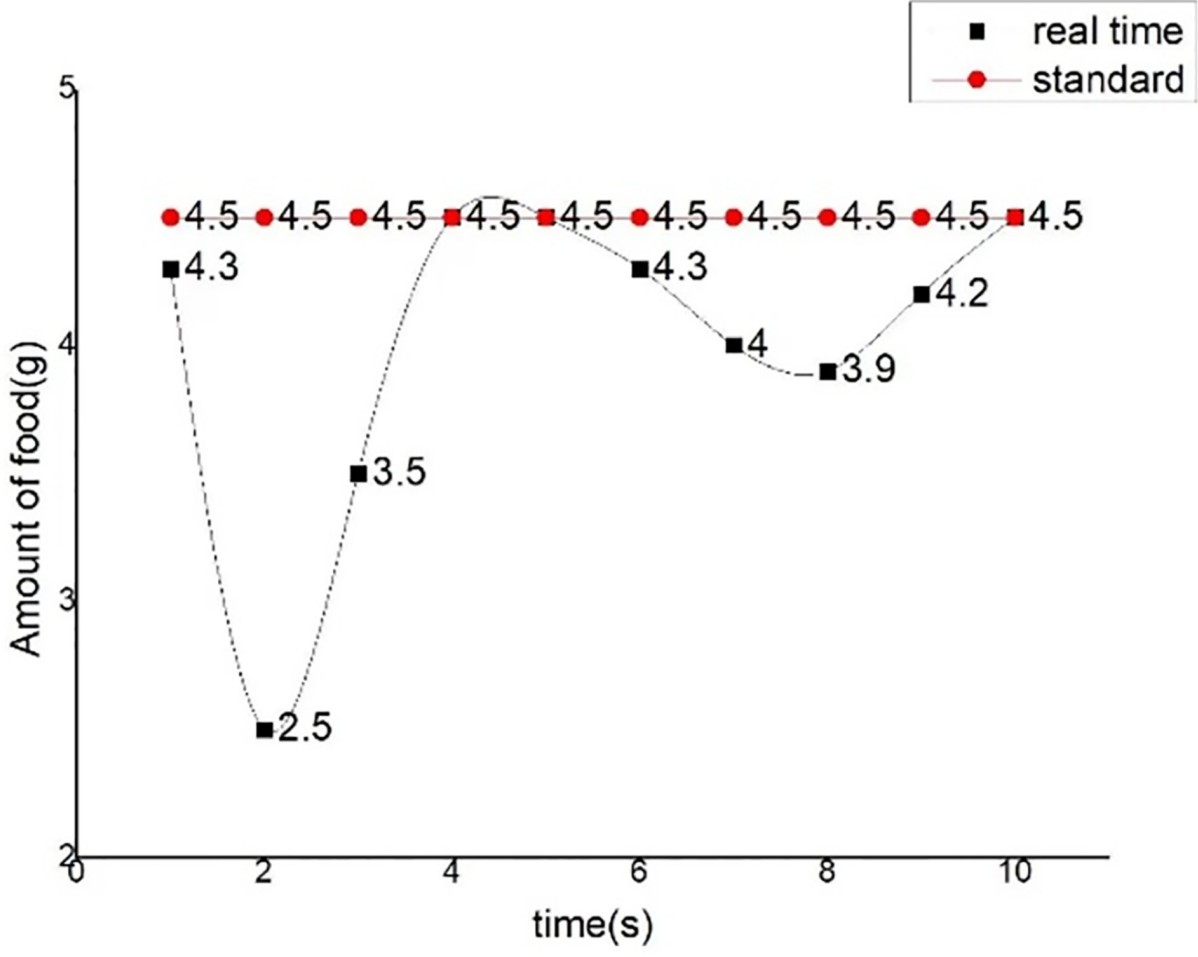

**Fig 25. Comparison chart of motor opening.**

## VI. Conclusions

This paper takes the real-time remote monitoring and management of the intelligent fish tank system for the purpose, according to the convenience of providing users and reducing the difficulty of fish farming as the starting point, the proposed intelligent fish tank remote monitoring and control system, after testing, the system installed all kinds of sensors can work normally, can monitor the environmental status of the fish tank in real time and report to the cloud, users can remotely view environmental parameter information through the APP, and can also use the functions on the APP for remote feeding, water purification, oxygenation operations, Users can also use the automation mode to let the fish tank be automatically managed, which has been tested and verified to operate efficiently, accurately and stably, bringing convenience to user management and saving time.

However, due to the current general price increase of chips on the market, in order to save costs, a more suitable main control chip will be studied, and this experiment is tested in WIFI mode, there are certain limitations, with the establishment of a large number of 5G base stations in the country, in order to further improve the stability of the network, will study the use of 5G network to replace WIFI mode.

## Author Contributions

**Conceptualization:** Shaohua Fu.

**Data curation:** Shaohua Fu, Jiangang Chen.

**Formal analysis:** Shaohua Fu, Wenjing Xing.

**Methodology:** Wenjing Xing.

**Project administration:** Shuangjiao Liu.

**Resources:** Wenjing Xing, Shuangjiao Liu.

**Software:** Juncheng Wu.

**Supervision:** Jiangang Chen.

**Validation:** Juncheng Wu.

**Writing – original draft:** Wenjing Xing.

**Writing – review & editing:** Wenjing Xing.

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
