## [Decision Letter · Decision Letter 0]

13 Mar 2023

PONE-D-22-31227Research and design of an intelligent fish tank systemPLOS ONE

Dear Dr. fu,

Thank you for submitting your manuscript to PLOS ONE. After careful consideration, we feel that it has merit but does not fully meet PLOS ONE’s publication criteria as it currently stands. Therefore, we invite you to submit a revised version of the manuscript that addresses the points raised during the review process.

We look forward to receiving your revised manuscript.

Kind regards,

Viacheslav Kovtun, Dr.Sc., Ph.D.

Academic Editor

PLOS ONE

Journal Requirements:

NO

This research was supported by the project fund of 2022 Science and Technology Project of Chongqing Municipal Education Commission "Research on the Design of Smart Farm Sharing Platform and the Optimization Method of Network Virtual Force Coverage" Fund no. KJQN20203419; In 2021, Chongqing higher vocational education teaching reform research project "Crafty Dream  Discussion on Ideological and Political Reform of  Embedded  Programming" ; Chongqing Education Commission Group Innovation: Self-Driving Vehicles Driving Together, Fund no. CXQT21032;

NO

NO authors have competing interests

6. Please amend the manuscript submission data (via Edit Submission) to include author Juncheng Wu ,Jiangang Chen,Junlong Huang,shuangjiao Liu, Wenjing Xing.

7. Please amend your list of authors on the manuscript to ensure that each author is linked to an affiliation. Authors’ affiliations should reflect the institution where the work was done (if authors moved subsequently, you can also list the new affiliation stating “current affiliation:….” as necessary).

8. Please remove your figures from within your manuscript file, leaving only the individual TIFF/EPS image files, uploaded separately. These will be automatically included in the reviewers’ PDF.

Reviewers' comments:

Reviewer's Responses to Questions

**Comments to the Author**

1. Is the manuscript technically sound, and do the data support the conclusions?

Reviewer #1: Yes

Reviewer #2: Yes

2. Has the statistical analysis been performed appropriately and rigorously? 

Reviewer #1: N/A

Reviewer #2: Yes

3. Have the authors made all data underlying the findings in their manuscript fully available?

Reviewer #1: Yes

Reviewer #2: Yes

4. Is the manuscript presented in an intelligible fashion and written in standard English?

Reviewer #1: Yes

Reviewer #2: Yes

5. Review Comments to the Author

Reviewer #1: I appreciate the fact that from research a developed product comes out. However, the research component is one of the most important aspects of the product that is required in the research community; as a result, the development of the "fish tank system" that research proposed is missing a significant amount of the research components, which can be summarized as follows:

• While there is theoretical foundation of the algorithm of the data collected by the sensor, and a good description of an improved first-order lag average filtering algorithm, yet there is a need for the provision of the practical applications scenarios for the implementations, specifically in SECTION 4.

• While there is theoretical foundation for the system's usage of composite collection information, intelligent processing, chart data analysis, there practical implication from the finding of the study was not highlighted

• The end of the smart fish tank that deals with remote monitoring and control, as well as a visual data interface, lacks an in-depth analysis for the support of your development

Reviewer #2: The paper presents a real time monitoring fish tank system. In order to support the research on the designed system, it is recommended to the authors to include a summary table after the conducted research. While there is strong motivation, the performance issue highlighted is difficult to understand. Some paragraphs need to be improved with the correct conjunction. English improvement should be carried out before the final version submission.

6. PLOS authors have the option to publish the peer review history of their article (what does this mean?). If published, this will include your full peer review and any attached files.

Reviewer #1: No

Reviewer #2: No

---

## [Author Response · Author response to Decision Letter 0]

30 Mar 2023

Reviewer #1:First of all, thank you very much for your suggestions, and secondly, we have made the following changes based on your suggestions. The actual use scenario of this design is the indoor fish tank equipment, and the improved first-order lag average filtering algorithm is used in this equipment system. We use different algorithms to calculate the data samples collected in real time, and make their comparison tables and graphs in Section 4 of the manuscript，the comparison results show that the improved first-order lag average filtering algorithm has superiority. For the compound information collection method, we refer to the relevant literature in the introduction part, analyze its advantages and disadvantages, and improve the method that is beneficial to this system. For the intelligent processing method, we use improved algorithms and better-performing chips and sensors to process data in experiments. For the graph data analysis method, we draw graphs in Section 4 and 5 to compare data samples. By strengthening user feedback and surveys, we establish a close relationship with users of smart fish tanks and visual data interfaces, and collect user feedback and needs so that we can better understand their needs and develop new functions and improvements.

Reviewer #2:Thank you for your suggestions on this article. According to your suggestions, we have made a hardware selection summary table at the end of section 3, and its parameters include the current and voltage of the hardware.

---

## [Decision Letter · Decision Letter 1]

17 Apr 2023

Research and design of an intelligent fish tank system

PONE-D-22-31227R1

Dear Dr. fu,

We’re pleased to inform you that your manuscript has been judged scientifically suitable for publication and will be formally accepted for publication once it meets all outstanding technical requirements.

Kind regards,

Viacheslav Kovtun, Dr.Sc., Ph.D.

Academic Editor

PLOS ONE

Additional Editor Comments (optional):

Reviewers' comments:

Reviewer's Responses to Questions

**Comments to the Author**

1. If the authors have adequately addressed your comments raised in a previous round of review and you feel that this manuscript is now acceptable for publication, you may indicate that here to bypass the “Comments to the Author” section, enter your conflict of interest statement in the “Confidential to Editor” section, and submit your "Accept" recommendation.

Reviewer #2: All comments have been addressed

2. Is the manuscript technically sound, and do the data support the conclusions?

Reviewer #2: Yes

3. Has the statistical analysis been performed appropriately and rigorously? 

Reviewer #2: Yes

4. Have the authors made all data underlying the findings in their manuscript fully available?

Reviewer #2: Yes

5. Is the manuscript presented in an intelligible fashion and written in standard English?

Reviewer #2: Yes

6. Review Comments to the Author

Reviewer #2: I recommend accepting the journal paper because the authors have addressed all of the reviewers' comments and suggestions. The revisions have improved the paper's quality and contribution to the field. The authors have demonstrated scholarly rigor, and I am confident that the paper will be of interest to researchers and practitioners

7. PLOS authors have the option to publish the peer review history of their article (what does this mean?). If published, this will include your full peer review and any attached files.

Reviewer #2: No

---

## [Editor Report · Acceptance letter]

26 Apr 2023

PONE-D-22-31227R1 

Research and design of an intelligent fish tank system 

Dear Dr. Fu:

I'm pleased to inform you that your manuscript has been deemed suitable for publication in PLOS ONE. Congratulations! Your manuscript is now with our production department. 

Kind regards, 

on behalf of

Professor Viacheslav Kovtun 

Academic Editor

PLOS ONE